# ‘*There Is Not a Word*’, but Is It Necessary? Analyzing Pragmatic Decisions Regarding Terminology Within Multispecies Family Relationships

**DOI:** 10.3390/ani15040568

**Published:** 2025-02-16

**Authors:** Javier López-Cepero, Alicia Español, Ángel Rodríguez-Banda

**Affiliations:** 1Departamento de Personalidad, Evaluación y Tratamiento Psicológicos, Facultad de Psicología, Universidad de Sevilla, 41018 Sevilla, Spain; aroban01@gmail.com; 2Departamento de Psicología Experimental-Psicología Básica, Facultad de Psicologia, Universidad de Sevilla, 41018 Sevilla, Spain; aespanol@us.es

**Keywords:** pragmatics on human communication, multispecies family, human–animal bond, pet parenting, companion animals

## Abstract

This study analyzes how people who live with companion animals decide which terms to use to talk about their animals. Increasingly, companion animals are considered part of the family, even as *furry children*, but how is this reflected in our language? This study was carried out through group interviews and the analysis showed that the labels are chosen based on two questions: what the animal means to the family and how much support they find in the social context. The results show that animals can play an important role in family life, but social pressure can lead to hiding the relationship, masking it through jokes, or even isolating oneself to prevent hostile comments. These results point to the challenges that these families face in their context and encourage us to analyze how the social environment can hinder their adaptation and visibility.

## 1. Introduction

In Spain, the presence of nonhuman animals within the family home has grown steadily in recent years, and it is estimated that 40% of households have at least one companion animal [1]. This increase in physical presence has been accompanied by changes in the status recognized for companion animals (i.e., those that live at home with the main function of providing companionship to the humans with whom they cohabit), being considered as another member within the family by more and more people [2,3].

These sociological changes are happening at great speed [4], fostering the appearance of new discourses about human–animal relationships. Some authors proclaim that we are facing an animal turn in our societies, characterized by a greater interest in their welfare—including the creation of specific legislation (Law 7/2023, March 29th, for the Protection of Animals’ Rights and Welfare) [5]—and the recognition of a higher social status [6]. However, these new discourses coexist with more traditional options, which facilitates the emergence of controversies and debates not yet overcome.

The term anthrozoology delimits an area of study focused on the interaction between humans and other animals, combining contributions from biology, psychology, sociology, veterinary science, and many other disciplines [7]. Anthrozoological studies show that our relationship with nonhuman animals faces inconsistencies and dilemmas that are difficult to solve [8], and animal labeling is an object of study where this is clearly reflected.

From a general point of view, it is easy to corroborate that the bulk of the population does not usually use scientific criteria when labeling the animals around them, but rather they tend to use category systems based on specific cultural traditions. These *folk taxonomies* [9,10] are efficient and help to guide the functioning of communities, but often involve inconsistencies. For example, the classification of animals into *pet*, *pest*, and *profit* [11] provides a simple classification system, but can lead to pragmatic paradoxes; thus, most of the Spanish population classifies cats and dogs as *pets*, but a small percentage (0.4% and 2.4%, respectively) consider them *pests*; in addition, while most participants qualify rats as *pests*, a quarter of participants consider them to be a *pet* and, more interestingly, a percentage of participants considered both options to be correct (rats were *pets* and *pests* at the same time) [12]. These inconsistencies are not exclusive to Spain, as cross-cultural studies point to the transversality of the phenomenon [13,14].

These phenomena have also been studied in more concrete situations, focusing on the labeling of specific individuals. The choice (or nonchoice) of names to refer to individuals has a close relationship to our consideration of their status and rights [15]. Naming nonhuman animals implies a recognition of their selfhood, of their existence as individuals, and leads to considering their self-experiences (including agency, coherence, affectivity, and history) [16,17]. Several authors have pointed out that researchers often avoid giving individual names to animals *used* in experiments as a means of reducing empathizing with these individuals, who are often destined to die at the end of their participation [18,19]. Therefore, the choice of terms does not derive directly from the qualities of the animals; it depends more on what relationship we want to establish with them and on the interaction we aspire to create.

In Spain, there are several options for naming companion animals. On the one hand, we find options characterized by the assimilating of solutions created to speak of other relationships. Previous studies describe some of these labels used, including terms developed to talk about human relationships such as ‘family members’, ‘children’, and ‘friends’ [20]. Other authors have proposed the use of neologisms that combine the human relationship with references to nonhuman animals, such as ‘furry babies’ [21,22]. From a quantitative point of view, most of these works assimilate the role of companion animals to that of offspring. Laurent-Simpson [23] speaks of ‘pet parents’, a term that reflects the identity of those people who recognize themselves in the role of raising these animals, including care, self-sacrifice, and even becoming financially indebted or refusing to evacuate in emergencies if they do not have shelters that can house them [24,25]. Several studies point to a greater tendency to treat and name animals as children in families that do not have human offspring [2,26].

On the other hand, some authors have incorporated brand-new terms to accommodate the new family organizations, such as ‘more-than-human families’ [3] or ‘multispecies families’ [27]. These terms attempt to reflect the importance of the interspecies link but also to overcome an anthropocentric bias: while traditional terms demonstrate concern for animal welfare, they also convey that recognition of their status depends largely on their humanization. The dictionary of the Royal Academy of the Spanish Language defines humanization as ’to make someone or something human, familiar and affable’, a definition that runs closer to the concept of anthropomorphizing (recognizing human qualities in nonhuman animals) [28].

Within anthrozoological studies, humanization has been defined in different ways. Beyond the use of categorial differences (e.g., pet vs. no pet), some authors have proposed dimensional approaches to this concept. First, Shir-Vertesh [29] analyzes humanization as a bipolar dimension, according to which animals see their rights recognized to the extent that they are assimilated to—or approximate—human beings: the greater the humanization, the greater the interest in guaranteeing their welfare. Second, López-Cepero and Español [30] propose a two-dimensional model that analyzes humanization as a plane with two axes: the level of bonding to the animal and the level of similarity (or difference) in needs between species. Unidimensional models imply a paradox, since higher humanization may entail high interest in animal wellbeing at the same time that it leads us to ignore ethological differences between species, thus obliging the companion animals to live a life that does not match their needs [6]. Despite being a term widely used in anthrozoology, the definition of humanization is still under debate [8].

Therefore, it is not only a matter of locating a term that denotes the incorporation of the animal into the family nucleus, but also of paying attention to the connotations of the term [31,32]. The use of conventional labels—or neologisms derived from them—seems an efficient solution, since it evokes relationships that all interlocutors are familiar with; however, these options may be met with rejection when the environment does not recognize the importance of the bond between humans and other animals (e.g., the use of terms such as *owner* or *pet*, which are increasingly rejected by the population for connoting a relationship of domination, although buying animals is common); [20,33] or when they are perceived as an attack on socially accepted family organization schemes (e.g., when people decide to live childfree and cohabitate with companion animals) [34,35].

Finally, it is also worth thinking that the context will influence the choice of one term or another to refer to the animal. Some works have drawn attention to the fact that people who live with companion animals tend to use labels such as parent when speaking in front of familiar people but tend to opt more for labels such as owner when in front of coworkers or strangers [2,36]. Thus, it appears that social context plays a role in term choice or use, although the mechanisms by which they affect the pragmatic decision have not been attended to so far.

Given this background, the present work aims to explore by means of qualitative methodology the terms used to name companion animals. The objective is not limited to the descriptive aspect—knowing the terms used—but seeks to analyze the decision-making process—balancing identity and adaptation to the context—underlying this labeling. The aim is to answer the question, is there a term capable of putting an end to the existing debates?

## 2. Materials and Methods

*Participants.* Three focus groups were conducted, with the participation of a total of 11 women and 1 man. All were between 18 and 64 years old, most were from urban contexts, and reported having lived with at least one companion animal for at least one year of their lives. Most of them lived in multispecies families that they had created with their partners, although three participants still lived at home with their parents. Table 1 shows the main sociodemographic data and the distribution of participants per focus group. Participants were selected based on the information included in an online form. The selection criteria were that the participants were Spanish speakers, of legal age, and had cohabited with an animal in the six months prior to the interview. In addition, the relationship with the animal had to be limited to domestic cohabitation, excluding those cases in which the relationship with the animal was based on work, such as veterinary care.

There is no consensus on the minimum number of participants or focus groups for qualitative studies [37]. However, the number of focus groups conducted in this work is sufficient to address the study aim of being a medium-sized project [38]. Likewise, the aim of the study and the specificity of the sample, among other elements, have provided the interviews with adequate information power, making a reduced number of focus groups necessary [39]. Implications derived from the sampling methods are analyzed in the discussion.

*Instruments.* A semi-structured ad hoc interview was created. Three main themes were addressed, each consisting of several questions: (1) the relative position of the animal within the family (e.g., *We would like to know, at home, how do you refer to the animals you live with?*); (2) the functional organization of the family (e.g., *What do you do together on a daily basis? How do you organize your day?*); and (3) the assimilation of animals to human relational categories (e.g., *If there is a change in the family* (e.g., *a birth), can the consideration of the animal change? In what way?*).

The focus groups were conducted virtually, through the BlackBoard Collaborate platform (Class Technologies, Washington DC, USA), in private, invitation-only sessions. This platform made it possible to record the audio of the conversations held.

*Procedures.* Online recruitment was carried out through the institutional profiles of the main researcher and the [research group] on the Facebook and Twitter social networks. Following the requirements of the ethics committee, participants filled out a form that contained informed consent. The participants were distributed in the different focus groups according to their availability.

The participants needed access to the Internet and an electronic device to conduct the interview. Following the privacy recommendations of Lobe et al. [40], participants were invited to use pseudonyms to identify themselves, as well as to decide the degree of exposure they wanted during the interview, choosing to share only the audio or also the image. We conducted online focus groups [41], thus facilitating the participation of people from different parts of the Spanish territory.

The focus group meetings lasted an average of 103 min. They were recorded and transcribed verbatim, attending to semantic content [38]. They were analyzed following the thematic analysis method proposed by Braun and Clarke [38,42]. The analysis was developed following six stages, from the identification of concrete codes up to proposing broad and abstract themes, going through several stages of refinement and revision. The three authors performed the coding with at least two of them always present, using NVivo software, Version 12 (Lumivero, Denver, CO, USA).

The results of the entire project yielded four overarching themes: (1) the creation of the relationship; (2) the choice of terms used to name the relationship; (3) the life trajectory of the multispecies family; and (4) the organizational schemes of the multispecies family. This article presents the results of the second overarching theme, reserving the explanation of the others for other publications.

## 3. Results

Participants used dozens of terms to refer to their relationship with their animals, including conventional options (*pet-owner*, *companion animal*), assimilations of human bonding terms (*friends*, *parent–child*, or neologisms such as *dogthers*—*perrhijo*, in Spanish) and novel terms (e.g., *multispecies family*). However, the present study is not interested in the enumeration of terms or their relative frequency, but rather in understanding what makes the participants prefer the use of one or the other in a given context. The choice of term does not seem to be static but depends on various discursive strategies.

The analysis process allowed us to agree on a model composed of three interrelated themes, which encompassed the following: (1) *What you mean to me*, which contemplates different types of human–animal relationships; (2) *Others’ surveillance*, which encompasses the role of social pressure in decision making; and (3) *A good solution (here and now)*, focused on the strategic decisions made to balance the above conditions. The present study focuses on the latter (*A good solution here and now*), and Figure 1 shows the thematic map resulting from the analysis carryed our for the current paper.

### 3.1. What You Mean to Me: A Combination of Humanization and Attachment

The analysis of the discourses offered by the participants made it possible to establish a first theme, the result of combining two variables: level of humanization and attachment to the animal. *Humanization* here does not refer to anthropomorphizing (recognizing human qualities in nonhuman subjects); it rather refers to the level of inclusion of nonhuman animals in human routines and roles that are not natural to them. When the level of humanization is high, companion animals may participate in family conflicts (e.g., being included in the triangular dynamics that may arise in family conflicts), or be considered friends, children, or other roles that humans usually play [43,44]. At the opposite pole of the dimension, a low level of humanization implies that companion animals would be considered categorically different from humans, thus being treated ‘as animals’. The following paragraph shows an example of how companions may be included in human habits and events, even when they do not make profit from them.


*[On celebrating companion animals’ birthday] I think we do this for us, for humans, rather than for them. They will enjoy that there will be a cake, a special can of food, that we make every effort to pay attention to them, because they love it when we pay attention to them. But I don’t think they understand that it is their birthday or that it is their adoption date, I don’t think it makes them any happier than what we already make them on a day-to-day basis, but it is something else for us.*
[Focus Group 2, MG]

*Attachment* refers to the intensity of the bond created with companion animals. It can be referred to as a phenomenological experience—e.g., how strong the participants feel the relationship is—or it can be expressed in terms of centrality—e.g., the importance of the animals in daily decision making and the search for their comfort [16,45].


*A few years ago, I was living abroad for a few months and I couldn’t get used to being without my cats and my dog […] all my life we have had animals at home and, I don’t know if it is because of that, but I cannot imagine it, because I don’t think I would ever be able to get rid of any of my animals.*
[FG3; EL]

The combination of both variables made it possible to abstract four profiles of the incorporation of companion animals into the families (see Figure 2). These profiles should not be understood as a rigid categorization, since discourses on the meaning of the relationship usually mix elements located at different levels of humanization and attachment, generating readings full of nuances similar to other anthrozoological studies [8]. Examples of the four profiles, as well as the most frequently used labels, are given below.

#### 3.1.1. Discourses That Assimilate Parent–Offspring and Other Human Relationships

Participants used labels that refer to motherhood as a reflection of the high humanization and bonding that shapes this relationship. Discussions about the use of terms such as mother/father, children, or other assimilative neologisms (pet parenting, cross-species parenting, furry babies, etc.) were similar to those described in previous studies [2,23,36], denoting that it is a very frequent solution, at least in industrialized Western countries.

What justifies the assimilation of these terms? The participants highlight elements such as the proximity between species, on a cognitive and emotional level, in addition to recognizing the dependence of companion animals on the human responsible for their wellbeing. Companion animals are perceived as dependent as infants are on adults. Participants feel that caring for their animals is a task comparable to caring for a baby. They feel affection and tenderness for their animals and assume responsibility for their care and the obligation to attend to their needs. This is how they experience and understand that these are similar feelings to those that mothers have for their children.


*As for calling an animal a child […] I understand that in many people’s heads it grates and that is why there are many discussions on social networks and so on about ’how can you equate a dog with your child’. And maybe those of us who do not have children are the closest thing we have to that responsibility, to that feeling of being a father and a mother […] Obviously the needs of a girl are not going to be the same as those of the dog, but the love is unconditional towards both, and I understand it that way.*
[FG 1; MR]

In most cases, the use of these terms was accompanied by a series of nuances and limitations. This shows that despite being the most common option among participants, the assimilation of the relationship to parenthood is not entirely satisfactory in a number of cases. Even if the experience is close, the fit is not perfect.


*Well, I think of her as if she were my daughter, because it is as if I were an adoptive mother. I have not given birth because she is not my species, but all the care I offer her, I try to make her as comfortable as possible and that she enjoys the life she has in the family… I consider it or I have the feeling that she is my daughter. […] I am not a mother, but I am a dog’s mother because I look after her safety, her wellbeing, I cover all her needs if she gets sick, if anything happens to her.*
[FG 1; BE]

Mentions of the parent–offspring bond were the clearest example of assimilation of human relationships, but not the only one. References to friendship or companionship between individuals of both species also appeared. These terms (*friend*, *companion*) imply both mutuality in the relationship and equality between those involved, which does not fit with some of the duties of caregivers —e.g., determining schedules, feeding, or preventing them from wandering freely in or out of the home.


*Well, I want to see them as companions, but with a very particular companion [laughs], because I think cats are more at home than outside, aren’t they? They are like home companions, I think they would place them on the level of friendship, in a certain way.*
[FG 3; CR]

#### 3.1.2. Discourses That Emphasize the Multispecies Composition of the Family

Discourses that assimilate terms proper to parenthood or friendship denote a strong bond of attachment and high humanization, giving companion animals a status close—if not identical—to humans. However, this attribution implies that animals, insofar as they are considered important, are added to schemes of human functioning (or habits), which may imply less respect for their ethology and individual needs.


*I think it is very different. I don’t have children and I don’t know how it feels, but I think that yes, it is different, that you cannot humanize a dog either, can you? […] I think the care and other things can be similar, the responsibility you have with him can be similar to the one you have with a child. In the end, they both depend on you, but I think you have to separate a human from an animal. Even if you treat it many times in the same way, even if you can be very fond of them, I think it is something different.*
[FG2; MJ]

In the interviews, mentions of the discomfort of assimilating interspecies relationships to human relationships were frequent. Here, new-fangled terms come into play, such as *multispecies family*, *more-than-human family*, or *posthuman family* [22,46]. Those labels attempt to reflect the intrinsic differences between the species while being respectful of the particular needs of each family member. In Spain, these terms have gained visibility in the last decade but are far from being used by most of the population.

#### 3.1.3. Discourses That Depict Animals as Objects of Prestige or Fashion

A third category of relationships was characterized by high humanization—in the sense that animals are incorporated into human dynamics—and weak attachment, where respect for the needs of the companion animal does not appear as a priority. In these cases, terms of frequent and shared use were not located, but animals were characterized as *objects* or *goods* of utility for humans—for example, as exclusive elements that symbolize prestige or fashion [47]. They can be defined by the roles they play—such as *influencer* dogs, understood as those that are used to monetize content on the Internet—more than by their selfhood (i.e., agency and other indicators of selfhood) [16].


*I know people who have tried to make their dogs influencers. They disguise them, publish the letter of the hundred, the ten thousand followers, all this. But well, I think that has another objective, not so much the feeling of union but trying to monetize your relationship with the animals, but that is my point of view.*
[FG 1; MR]

The participants agreed to point out these relationships as ethically controversial. In addition, dilemmas about the use and objectification of animals intersected with other ongoing debates, such as the ethics of purchasing—rather than adopting—companion animals or the risks that the selective breeding of certain fashionable breeds poses to animal health.


*I remember when I adopted my dog. He was one month old, and when we brought him to my parents’ house a friend came by the door and I was carrying him in the carrier. And then he stopped in the car and he said ‘let’s see him, let’s see him’, and I showed the dog to him. And he said ‘what is this?’, I said ‘a dog’, and he said ‘ah, that’s a mutt’, and it hurt my soul […] And sometimes I feel like an attack. Why does a dog have to be ‘the best’?*
[FG 2; MG]

#### 3.1.4. Discourses That Qualify Animals as Pets of the Family

Finally, a group of discourses is characterized by a weak attachment and the perception of discontinuity between species. In these cases, the animal appears as a complement to the true nucleus of the family, composed of humans, establishing an unequal relationship in which humans are placed above other species in terms of status.

The terms used within these discourses included the *pet-owner* binomial and the use of the species of the animal as a label, with mention of its belonging to the family (e.g., *the house dog* or *the family cat*). It is worth mentioning that the participants agreed that these terms, although correct in their denotative aspect, have acquired anthropocentric and negative connotations, in accordance with the previous literature [48]. Thus, many of the classic discourses shared by a large part of the population may be offensive to a growing percentage of Spaniards.


*MR: I don’t use the word pet, for example. If I have to refer to my cat, I say ‘my cat’.*



*LU: Exactly, ’your cat’. I think what I mean, for me, is to say, for example, ’pet’ or ‘animal’ is like saying to your mother ‘that person you know’. Or to someone who is in your family to say ‘that person’, ‘the people in my house’. [smiles]*
[FG 1; MR and LU]

As a final note, the terms *companion* and *companion animal* were described on several occasions as labels that could fit within this category. This finding is not necessarily representative of the general Spanish population, but points to a phenomenon of interest: although there are alternatives to the term *pet* and they denote a certain equality between the parties, some participants considered it an ‘old-fashioned’ and ‘impersonal’ term. This is one more sign of how quickly language use in this area is changing.

### 3.2. Others’ Surveillance: Who Am I Talking to?

The previous theme depicted four types of human–animal relationship, but it only represented the first component of the equation when it comes to selecting the most appropriate term. The second theme refers to the social pressure perceived by participants when talking about their animals. The social context plays a moderating role in the choice of terms, causing participants to use different choices depending on the context in which they are made.

The role of the social environment in the description of bonding has been explored in previous work, where the impact of cultural differences is noted [36]. Our participants emphasize that the intimacy maintained with the interlocutor determines the degree of openness in showing the real importance of the human–animal bond in the discourse. In a similar way to that described by Volsche [2], conversations held with coworkers or strangers tend to include less personal references to companion animals, while labels referring to parenthood or familiarity were more present in conversations of greater intimacy. This pressure felt by observers may lose relevance as the participant evolves, as if it were a conquest.


*With my family and friends, I always called him ’my boy’, ’my baby’, always something affectionate or nicknames I gave him, depending on the moment, I would say one thing or another. But it is true that when there were people around me who were prejudiced, who used to say ’he is a dog, he is not a person’, then maybe I was more self-conscious. […] But as time went by, it is true that I did not care who was in front of me, if it was important to me and I wanted to say ’my child’ or ’my baby’ or whatever I had to say, I said it […] I became less and less concerned about the opinions of people.*
[FG 2; MJ]

The moderating role of social context and perceived social pressure is explored in the next theme, which discusses concrete examples of how participants made decisions about which labels to use. How do you decide which term is more appropriate to refer to companion animals in each situation?

### 3.3. A Good Solution (Here and Now): Taking Pragmatic Decisions

Therefore, participants report that the choice of terms is related to the meaning of the bond and the context in which it is used. But how do these two issues come together to make pragmatic decisions? In this third theme, we explore the strategic decision process that makes it possible to maintain balance between all the aspects involved. Following the axioms proposed by the theory of human communication [31,32], we assume that it is impossible not to communicate, and that every communication sends multiple messages referring to both the content (i.e., what is verbally expressed) and the relationship (i.e., how I understand the situation) that create an impact in the ones that receive it. Schematically, we will present these options in three categories (Figure 3).

#### 3.3.1. Affirmative Strategies: ‘We Are as We Are’

The first strategy consists of publicly affirming the felt relationship. We refer to situations in which participants express in a normalized way the term they feel to be the best for describing their relationship. Of course, this presents an undemanding challenge to people who feel low attachment, as well as to those who conceive a clear discontinuity between humans and other animals—e.g., when the family *has* animals—as these statements coincide with traditional and socially accepted discourses [3,23].


*For me it is something completely different, the decision to have children or not, to be a mother or not […] for me it is completely different, and it does not replace or would not replace in the case of being a mother what a child is. In other words, I consider them an important part of my family, but it has nothing to do with motherhood, nor is it related to any of that.*
[FG 2; LS]

Affirmation strategies require more effort when the value of the relationship is high, but the social context is hostile to the assimilation of nonhuman animals as family. Affirmation may include the use of neologisms to adapt conventional labels or post-conventional terms, although both options present advantages and disadvantages. Several participants refer to the attrition this strategy entails in a manner similar to that described in previous work [34,35].


*I think in the end it is all going to generate controversy. If you have cats, if you don’t have cats. That is, if you have animal, if you do not have animal. If you consider her as a child, if not. Then, if it is going to generate controversy, because nowadays everything is going to generate controversy and everybody is offended, I think that ‘dogther’, ‘furry baby’ or kitten is more approximate because, being objective, she is an animal. But being subjective, she is your family. So of course, no one will understand it, only those who are in the same situation as you, or who at least feel the same way you do.*
[FG 1; LS]

In sum, affirmative strategies allow participants to express the importance of the bond in front of others. This affirmation implies personal costs, as the way they represent their relationship with companion animals may represent the risk of being socially criticised, especially when their construction of the bond may challenge the accepted speeches in their social context. Given that it is impossible not to communicate, debate is not avoidable [31,32].

#### 3.3.2. Denial Strategies: ‘Nothing to Declare (To You)’

A second strategy involved concealment or denial of the bond in an attempt to fit the social context. This occurs when the importance of the bond and/or the continuity between species is high, but the social pressure is too much for the participant. When participants anticipate that they will not be well received or understood, they avoid exposing their preferred terms, like those described by Vosche [35]. This was especially prevalent when the animal’s position was assimilated to the maternal–filial relationship (as reflected in the above quotation from participant LS). In these cases, withdrawal from the context or the search for safe spaces, such as Facebook groups or Twitter, seems to be a solution to compensate for the continuous denial of the relationship, which they can sometimes be forced to do.


*I express myself [in social networks] to see what interactions I get because many times people say ’I understand you’, ’I feel the same’, ’for me it’s like part of my family’ […] I think that social networks are also useful for that and, and it has been a step to realize that there are many people like you and that they live it or live it as you have lived it or are living it, right?*
[FG 2; MJ]

An alternative strategy does not imply physical withdrawal from the social context but hiding the importance of the bond. This is performed by actively avoiding the terms that determine the type of relationship. The use of proper names (e.g., Rex) or species names allows participants to avoid possible conflicts with the interlocutor.


*Basically, it depends on the receiver, but most of the time I refer to them with nicknames […] So if I am talking to my mother I say ‘the critters’. At home, they have their name, I say ‘the bulky one’ or whatever. Obviously, if it is a coworker I say ‘my dogs’.*
[FG 2; CR]

Denial strategies may be protective in front of others, but they oblige the participants to behave in a non-coherent way. Again, the impossibility of not communicating may lead the participants to speak as if the relationship was uninportant, an experience that can raise personal dilemmas and the sense of being disloyal with the companion animal.

#### 3.3.3. Mixed Strategies: Identity and Social Context in Equilibrium

Third and finally, participants reported several intermediate strategies in which the importance of the relationship was affirmed and then nuanced. For example, justifying the normality of their discourse, setting boundaries to define what would be absurd.


*[On speaking of dogs as children] It is because it is incorporated in our language, as they have a certain situation of dependence, that they depend on us in some sense, right? In the end it always comes out ‘watch out the girl’, but it is more like a term of endearment, it is not… because there are people who say ‘do not compare children with dogs, right?’ I do not confuse them, I don’t… I have not got there, yet [laughs].*
[FG 3; MC]

A second formula, frequently used, consisted of disavowal of one’s own position, qualifying it as a joke or ’way of talking’ [49,50]. In this way, they can denote the relationship as they feel it while controlling possible negative reactions that may occur in the social context. This type of solution bears resemblance to the idea that the use of generated terms for human families is nested in convenience, in linguistic thrift [7].


*I am very clear; for me they are completely different things. My animals are part of my family, but a son is a son, it is a completely different thing from my dogs. Even if I speak intimately, it is like a code, I don’t know, let me see if I can explain myself. When my partner and I say ‘go with mom or dad’, it’s like a joke code in a very intimate context, but we don’t both use it in a way, not seriously or anything. It is like a joke that we make to each other.*
[FG 2; LS]

## 4. Discussion

Companion animals occupy an important place in our societies and their growing popularity has changed the composition of Spanish households. Immersed in this sociological transition, the present study provides some insight into the process participants go through to choose the terms that reflect their identity as a family in different social contexts.

The first contribution of the present study is to challenge the idea that multispecies families need a single term to be described. The results show that families can include companion animals in many ways, depending on the level of humanization and the strength of the emotional bond established. The combination of both dimensions gives rise to a much richer theoretical plane than the one described in previous studies [20,29], allowing us to resolve paradoxical conceptualizations rooted in the field of human-animal interaction—for example, assuming that a more humanized treatment must imply a strong bond and an interest in their welfare, when it can also force the animal to live a denatured life [6].

A second finding allows us to highlight the friction and discomfort that can surround the choice of terms. Labels reflect family identity, but their expression is mediated by existing social pressure. Although this result is similar to previous publications [2,34], the present work provides a dimensional view of this variable; social pressure cannot be considered a single element but presents different intensities and varies among the various contexts (family, friendships, work, school, or neighborhood) of which participants are part.

Consequently, it seems unlikely that families can find a single term suitable for all contexts. The pragmatic approach to choices allows us to offer a more detailed view of the trade-offs made by participants in labeling the relationship they establish with their companion animals. On the one hand, by pointing out the strategic value of the choices; it depends not only on what the relationship means, but also on how I want to make it visible in a given context. On the other hand, navigating the difficulties inherent in the existence of various labels that not only have different denotations, but gain connotations with use. For example, classic terms such as *pet* and *owner* have gained negative connotations in certain communities, as they imply a clear discontinuity between species characteristics and human dominance [20]; however, they remain terms collected in most standards and laws of application in the field. In addition, many terms referring to nonhuman animals gain specific connotations regarding cultural context (e.g., colloquial uses). In summary, when one has no prior information about the dominant discourse in a context, anticipating the choice of the correct term is difficult.

These pragmatic difficulties intersect with issues that go beyond our relationship with companion animals. Diversity in family configurations has gained visibility in recent decades, but hegemonic social discourses may put extra pressure on people who have decided to live outside the most frequent models [34]. In Spanish society, families have traditionally been defined as social groups aimed at raising children, but recent studies show that the nuclear family configuration composed of heterosexual couple and children has not, for some time, represented the majority of Spanish households [4]. In this context, the rise of companion animals is often criticized, from traditional positions, for their role as surrogate children, thus questioning their validity as real relatives [23,34]. Therefore, although our participants clearly mark the existence of differences between living with human children and with companion animals, talking about their relationship in terms close to those of motherhood (i.e., caring for a dependent living being, which is considered as an individual with rights and aspirations, and with the vocation to offer them the maximum possible welfare) is a challenge with a difficult solution.

The strategies implemented by the participants partially coincide with what is predicted by the pragmatics of human communication [31,32]. First, when the relationship with the companion animal is important for the interviewee’s family identity, the most frequent choice corresponds to the affirmation of this relationship in the public sphere. However, even in these cases, there are indicators of conflict: the fragments analyzed show that it is not free of insecurities, and that it is usually described as an achievement obtained through a process developed over time. The second strategy consists of denying the importance of the bond and can be understood as an attempt not to communicate. The fragments analyzed show that participants use context avoidance and ellipsis—avoiding naming the relationship—as tools to avoid being confronted by the social context. Finally, the participants showed the existence of other strategies, such as the devaluation of the message itself. This is an intermediate case between the two previous ones, since it effectively names the existence of the relationship, to then mark the *reasonable* limits of this relationship or consider it a *joke*.

A significant part of the population considers companion animals to be full members of their families, but they are confronted with social discourses that often devalue their experience. Some of them are forced to escape from certain social contexts that they perceive as hostile, while others try to protect themselves from external criticism by placing their discourse in the realm of the false or the jocular. However, according to the axioms underpinning the pragmatics of human communication, the attempt not to communicate necessarily implies communication: not saying what you feel is a decision, and this decision may involve a conflict between the perception of your own family—your identity—and its representation in the public sphere.

Of course, not all people who live with animals face these dilemmas. As described in this paper, not all people construct their relationship with companion animals in the same way, and a significant part of the population does not aspire to form multispecies families. However, this work coincides with other voices pointing out the increasing importance of companion animals within families [16,23]. Therefore, it seems urgent to attend to the implications of these findings to give visibility and voice to multispecies families, helping them to integrate into the social fabric and to respond to their specific needs—for example, to study the option of taking sick leave to care for companion animals when they are ill.

The present study has limitations. Although the sample size is adequate for a thematic analysis in a medium-sized study [38], having 12 participants limits the options to represent all the discourses available in the social environment. Similarly, there was a majority of female participants, although this difference is similar to that described for other studies on human–animal interaction and can be understood as reflecting certain social values, according to which women tend to be more open to showing empathy and interest in animal care [51,52]. Furthermore, it would be advisable to improve diversity in future studies, as only one of the participants cared for other relatives at the time they participated in the study and most of them were under 40 years old. In order to learn more about the effects of the moment of the course of life on the human–animal bond, these variables should be specifically focused on. On the other hand, this study presents some strengths, as it included participants from both rural and urban settings, its use of video calls facilitated the participation of people from different parts of Spain, and it incorporated participants of different ages, with and without children, thus allowing some geographical and demographic diversity.

## 5. Conclusions

The conclusions derived from the present work are of interest to researchers and practitioners. From a scientific point of view, it is interesting to corroborate that some of the inconsistencies found in foreign studies also appear in the Spanish population. In addition, the present work provides novel theoretical elements, such as a two-dimensional model of the meaning of the human–animal relationship or the incorporation of the pragmatic theory of human communication, that can help to deepen our understanding of the bond generated between humans and other animals. Moreover, the applications of this knowledge are varied. In a world in which many people understand their bond with companion animals as just another family relationship, professionals in education, psychology, and many other disciplines must gain sensitivity to these new realities, making room for multispecies relationships when assessing the situation of families and designing interventions, both in micro (e.g., family therapy) or macrosystemic contexts (such as when designing family reconciliation policies), supporting the normalization of the animal turn to socially shared discourses.

## Figures and Tables

**Figure 1 animals-15-00568-f001:**
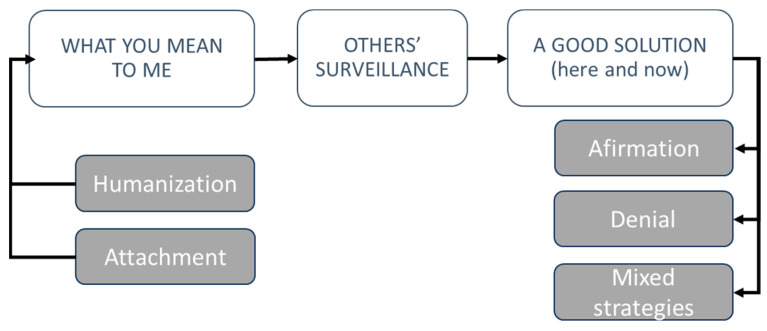
Thematic map derived from the thematic analysis for the present study.

**Figure 2 animals-15-00568-f002:**
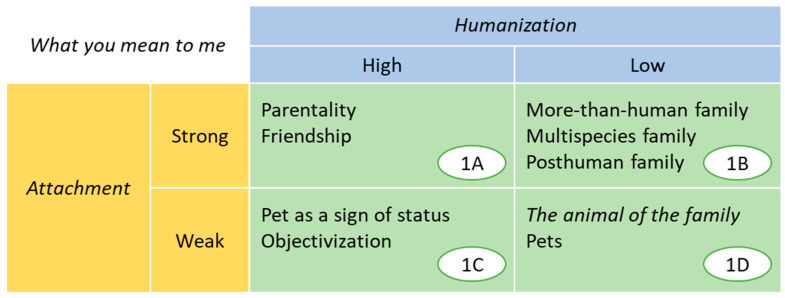
Four profiles of human–animal relations (from 1A to 1D) derived from thematic analysis.

**Figure 3 animals-15-00568-f003:**
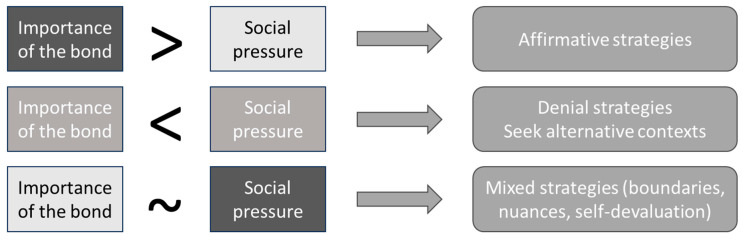
Decision making depends on the interplay between bond importance and social pressure.

**Table 1 animals-15-00568-t001:** Sociodemographic information of participants.

Focus Group	Tag	Gender	Age	Environment	Caring of Humans	Care of Animals	Current # of Comp. Animals	Species Cohabitated
1	CA	M	18–29	R	No	Shared	4–5	Dog, cat
	BE	F	30–39	U	No	Alone	1	Dog
	MR	F	30–39	U	No	Shared	1	Cat, other
	LU	F	30–39	U	Offspring	Shared	1	Dog, rabbit
2	CB *	F	30–39	U	No	Shared	2–3	Dog
	MG	F	18–29	R	No	Shared	1	Dog
	LS	F	40–64	R	No	Shared	2–3	Dog
	MJ	F	18–29	R	No	Shared	None	Dog, other
3	EL	F	18–29	U	No	Shared	2–3	Dog, cat
	RQ	F	18–29	U	No	Shared	None	Cat, other
	CR	F	30–39	U	No	Shared	4–5	Dog, cat
	MC	F	30–39	U	No	Shared	4–5	Dog

Gender: M = Man; F = Female. Environment: R = Rural; U = Urban. * Participant could not complete the interview due to technical problems.

## Data Availability

Data are available upon request.

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
