# Peer review of "‘*There Is Not a Word*’, but Is It Necessary? Analyzing Pragmatic Decisions Regarding Terminology Within Multispecies Family Relationships"

_animals, 2025, doi:10.3390/ani15040568_

Round 1

Reviewer 1 Report

Comments and Suggestions for Authors

I enjoyed reading this paper. It is mostly well written and structured but in places the way things are expressed makes it difficult to understand.

I do have a small number of concerns.

1. The authors state 'the aim of the study and the specificity of the sample, among other elements, have facilitated an adequate information power of the interviews, making a reduced number of focus groups necessary [39]'.

While I have no issue with small samples in this type of qualitative work there are some interesting consequences. The sample is mainly female (apart from one male participant) and this should give rise to some discussion of the gendered nature of language. Carol Gilligan's work - In a different voice offers useful insights into the difference between boys and girls as they  develop in respect of their relationship to moral decision making and to care, and the consequences of this for the language they use. In addition, the age profile of the participants was split between the 18-29yrs and the 30-39yr groups but according to the Table 1 only one participant cared for other humans. So this sample looks quite specialised and I would have liked to see some discussion from the authors about how this may have influenced the findings.

2. This paper focuses only on the second overarching theme - choice of terms to name the relationship. I did not really understand the thematic map in Fig 1.  This should be explained more clearly and should be linked to the purpose of the paper which is to explain how people actually make decisions about how they talk about their animals and the importance of context.

3. The use of language by the authors is often confusing - see the following examples

When the level of humanization is high, companions may participate in family conflicts or have consideration of friends, children, or other roles that humans usually play [43], [44]. page 6

How do companions participate in family conflict?

an example of how companions may be included in human costumes, even when they do not make profit of them. page 6

What are human costumes and what does make profit of them mean?

 Attachment refers to the intensity of the bond created with companions. It can be referred to as a phenomenological experience —e.g., how strong the participants feel the relationship. page 6

Attachment does refer to the relationship bonds but requires qualification to support ideas about intensity and I am not why or how phenomenology is being used in this context.

4. Concepts are introduced but it is difficult to see where they come from or exactly how they are defined. For example, Figure 2 Parentality / Post human family / Pets as a sign of status 

Discourses that assimilate terms proper to parenthood or friendship denote a strong bond of attachment and high humanization, giving companion animals a status close —if not identical— to humans. However, this attribution implies that animals, insofar as they are considered important, are added to schemes of human functioning, which may imply less respect for their ethology and individual needs. page7

What does this actually mean?

 Our participants emphasize that the intimacy maintained with the interlocutor determines the degree of importance given to the human-animal bond in the discourse page 9

What does this actually mean?

5. I think there are areas where more analytical work is required. In particular the development of the ideas contained within affirmative and denial strategies.

The discussion could be expanded to take account of the colloguial use of animal-related words within certain communities: In the UK the term 'pet' is used as a reference to other people (particularly in the North East of the UK), similarly 'duck' in Stoke on Trent in the UK.

Author Response

Dear reviewer,

We attach the responses to your concerns and comments in the document. 

Thank you so much for your insights. Kind regards,

Authors.

Reviewer 1

Comment

Response

I enjoyed reading this paper. It is mostly well written and structured but in places the way things are expressed makes it difficult to understand.

Thank you so much for your encouraging comment. We are glad that the manuscript was of interest. We will revise it thoughtfully to ensure that all the concerns are correctly addressed.

In the next comments, we have highlighted changes in red.

1. The authors state 'the aim of the study and the specificity of the sample, among other elements, have facilitated an adequate information power of the interviews, making a reduced number of focus groups necessary [39]'.

While I have no issue with small samples in this type of qualitative work there are some interesting consequences. (...) So this sample looks quite specialised and I would have liked to see some discussion from the authors about how this may have influenced the findings.

Thank you for the comment. In the paragraph you cited, we added:

“Implications derived from the sampling methods are analyzed in the discussion.  ”

Later, in the discussion > limitations, we added what follows (lines 492 and on):

“Furthermore, it would be advisable to improve diversity in future studies, as only one of the participants cared for other relatives at the time they participated in the study and most of them were under 40 years old. In order to learn more on the effects of the moment of the course of life on the human-animal bond, these variables should be specifically focused..”

We hope that it is better shape now.

2. This paper focuses only on the second overarching theme - choice of terms to name the relationship. I did not really understand the thematic map in Fig 1.  This should be explained more clearly and should be linked to the purpose of the paper which is to explain how people actually make decisions about how they talk about their animals and the importance of context.

We agree that it was poorly explained. This figure refers only to the present study, not to the whole project. We apologyze for the confusion.

In line 183 and on, we wrote the following:

“Present study focusses in the latest (A good solution here & now), and Figure 1 shows the thematic map resulting from the analysis carryed our for the current paper.”

In the footnote of the figure, now it says:

Figure 1. Thematic map derived from the thematic analysis for the present study”

3. The use of language by the authors is often confusing - see the following examples

When the level of humanization is high, companions may participate in family conflicts or have consideration of friends, children, or other roles that humans usually play [43], [44]. page 6

How do companions participate in family conflict?

We added an example, extracted from Leow’s and Walsh’s works:

“(e.g. being included in the triangular dynamics that may arise in family conflicts),”

We hope that it could help. We did not want to expand the paragraph too much. If further information is needed, please let us know.

an example of how companions may be included in human costumes, even when they do not make profit of them. page 6

What are human costumes and what does make profit of them mean?

We think that the term “costumes” could be misleading. We have changed it for “habits or events”, as we meant that animals can be included in contexts that are created for humans to enjoy, but that they cannot understand or enjoy (e.g. a party, where they cannot play or run as there are more people and less room to move). We hope that it is easier to read now.

Attachment refers to the intensity of the bond created with companions. It can be referred to as a phenomenological experience —e.g., how strong the participants feel the relationship. page 6

Attachment does refer to the relationship bonds but requires qualification to support ideas about intensity and I am not why or how phenomenology is being used in this context.

We referred to phenomenology as we speak of how the participants build up their understanding of the bond. We do not measure the bond observing their behavior (e.g. time spent together) or by physiological correlates; we only ask the participants how they experience the bond.

Perhaps we could refer to consciousness/awareness, but we are not sure if these concepts would help to improve the explanation. Please let us know if we have clarified this point.

4. Concepts are introduced but it is difficult to see where they come from or exactly how they are defined. For example, Figure 2 Parentality / Post human family / Pets as a sign of status 

In the paragraph before the figure 2, it says “examples are given below”. We have completed the footnote, that now says “four profiles of human-animal relations (...) derived from thematic analysis”. We hope that it is clearer now. Thank you for noting.

Discourses that assimilate terms proper to parenthood or friendship denote a strong bond of attachment and high humanization, giving companion animals a status close —if not identical— to humans. However, this attribution implies that animals, insofar as they are considered important, are added to schemes of human functioning, which may imply less respect for their ethology and individual needs. page7

What does this actually mean?

We acknowledge that the expression could be misleading, so we added a short note to clarify it. Now it is read:

“(...) are added to schemes of human functioning (or habits), which may imply less respect for their ethology”

We hope that it is clearer now. We are not native English-speakers, so we appreciate the question.

Our participants emphasize that the intimacy maintained with the interlocutor determines the degree of importance given to the human-animal bond in the discourse page 9

What does this actually mean?

We have reworded the phrase, that now is read (line 322):

“Our participants emphasize that the intimacy maintained with the interlocutor determines the degree of openness in showing the real importance of the human-animal bond in the discourse”

We hope that it is clearer now. Thanks for noting.

5. I think there are areas where more analytical work is required. In particular the development of the ideas contained within affirmative and denial strategies.

We have added the next paragraphs at the end of both sections:

Lines 374+ “In sum, affirmative strategies allow participants to express the importance of the bond in from of others. This affirmation implies personal costs, as the way they represent their relationship with companion animals may represent a risk of being socially criticised, especially when their construction of the bond may challenge the accepted speeches in their social context. Given that it is impossible not to communicate, debate is not avoidable [31, 32].”

Lines 399+: “Denial strategies may be protective in front of others, but they obligue the participants to behave in a non-coherent way. Again, the impossibility of not communicating may lead the participants to speak as if the relationship was uninportant, an experience that can arise personal dilemmas and the sense of being disloyal with the companion animal.”

We hope that those paragraphs add some ellaboration to these sections.

  The discussion could be expanded to take account of the colloguial use of animal-related words within certain communities: In the UK the term 'pet' is used as a reference to other people (particularly in the North East of the UK), similarly 'duck' in Stoke on Trent in the UK.

We have introduced a mention to this, around lines 448+:

“In addition, many terms referred to non-human animals gain specifical connotation regarding cultural context (e.g. colloquial uses). In summary, when one has no prior information about the dominant discourse in a context, anticipating the choice of the correct term is difficult.”

We hope that it will help the readers.  Thank you for noting.

Reviewer 2 Report

Comments and Suggestions for Authors

Interesting work, very well balanced, a real merit considering the topic

Only few remarks and suggestions

Throughout the text, please « Comapnion animals » contrary to only « companions »

L 71 Asessing that animals have « rights » is a real debate and should not be treated as a universal accepted fact / « Independent » awkward term, as many so called pets are actually captive animals.

L 93 Replace elements with animals

L99 delete « or attachments » - unnencessary repetition

L100 Better use similarity instead of proximity to oppose to difference

L108 Refusing owner is in contradiction with the fact that a large majority of pets are bought in shops or from breeders.

L124 Unclear. Whole sample =12 individuals or three times 12 ind. ? We later learn that it is 3 groups of 4 ind. each. Please clarify.

Fig. 1 Affirmation instead of afirmation

Author Response

Dear reviewer,

We attach the responses to your concerns and comments in the document. 

Thank you so much for your insights. Kind regards,

Authors.

Reviewer 2

Comment

Response

Interesting work, very well balanced, a real merit considering the topic

Thank you so much for this comment.

Throughout the text, please « Comapnion animals » contrary to only « companions »

We changed all the mentions to “companion animals”. We did not write these changes in red because there were around 30, but there are no mentions to “companions” remaining in the manuscript.

L 71  « Independent » awkward term, as many so called pets are actually captive animals.

We have removed the term “independent”, as it could be reiterative (individual includes the idea of individualization).

L 93 Replace elements with animals

We wanted to state that anthropomorphizing may take place also in objects (e.g. when we consider that the printer “does not want to obbey us”). But we changed it to refer to animals, given the scope of this study.

L99 delete « or attachments » - unnencessary repetition

Removed. Thank you for the comment.

L100 Better use similarity instead of proximity to oppose to difference

Changed. Thank you for noting it.

L108 Refusing owner is in contradiction with the fact that a large majority of pets are bought in shops or from breeders.

We have mentioned this idea:

(e.g., the use of terms such as owner or pet, which are increasingly rejected by the population for connoting a relationship of domination, although buying animals is common)

L124 Unclear. Whole sample =12 individuals or three times 12 ind. ? We later learn that it is 3 groups of 4 ind. each. Please clarify.

Now it can be read as:

“Three focus groups were conducted, with the participation of a total of 11 women and 1 man”

We hope that these changes match your expectations. Thank you so much for your feedback.

Reviewer 3 Report

Comments and Suggestions for Authors

Thank you so much for submitting this manuscript and adding to our understanding of human-animal relationships particularly in non-US UK populations. I was very impressed with the manuscript and am happy to accept the manuscript in its present form. I have two observations that are not crucial to the narrative of the text and hence not required for revision but if possible it would be wonderful to add to flesh out the narrative. 

Line 126: Do you have any insight if the amount of animals and/or if the participant considered themselves primary caretaker led to differences in their understanding of their relationship? 

Line 126: Did participants report different relationships between a dog or a cat even if in the same household? Or even between multiple animals of the same species. For example, one dog is more on par with "child" while another is not. 

Author Response

Dear reviewer,

We attach the responses to your concerns and comments in the document. 

Thank you so much for your insights. Kind regards,

Authors.

Reviewer 3

Comment

Response

Thank you so much for submitting this manuscript and adding to our understanding of human-animal relationships particularly in non-US UK populations. I was very impressed with the manuscript and am happy to accept the manuscript in its present form.

Thank you so much for your comment. We are glad that the manuscript is of interest.

Anthrozoology is growing in Spain and Spanish-speaking countries, and we hope to keep adding some insights to scientific literature.

Line 126: Do you have any insight if the amount of animals and/or if the participant considered themselves primary caretaker led to differences in their understanding of their relationship? 

Did participants report different relationships between a dog or a cat even if in the same household? Or even between multiple animals of the same species. For example, one dog is more on par with "child" while another is not. 

Great questions to comment.

We only had a participant that considered herself as the only carer of the companion animal, so we could not check if this variable could play any role.

When more than one companion animal cohabitated with the participants, no mention was made to differences in the way they relate to each one.

We think (we mean, it is an hypothesis that we could check in the future) that the lack of differences rely in the fact that most of the companions were mammals. Some of the participants mentioned that they could not relate “in the same way” with chickens or goldfish because they could not play with them, or they did not understand “what you want to tell them”. It would be of interest to learn more on the limits of these considerations, as horses and cows are mammals (intelligent, capable ones) but are not cohabiting at home. Would we find the same rigid differentiations, or would we rather find mixed speeches (they can understand, but still we eat/’use’ them)?

Thank you for the ideas, we are interested in keeping this line and we will definitely check those hypotheses in the future.